# Influence of Strategic Crisis Communication on Public Perceptions during Public Health Crises: Insights from YouTube Chinese Media

**DOI:** 10.3390/bs14020091

**Published:** 2024-01-26

**Authors:** Dan Sun, Yiping Li

**Affiliations:** 1School of Economics, Hunan Institute of Engineering, Xiangtan 411104, China; sundan@hnie.edu.cn; 2School of International Relations, Xiamen University, Xiamen 361005, China

**Keywords:** crisis communication, public perceptions, situational crisis communication theory (SCCT), sentiment analysis, public health crisis

## Abstract

Crisis communication plays a crucial role in preserving the national reputation during significant national crises. From the perspective of Situational Crisis Communication Theory (SCCT), this research paper analyzed over 1,790,816 YouTube comments from Chinese-speaking audiences, using sentiment analysis alongside the Difference-in-Differences (DiD) model, in order to investigate the influence of strategic crisis communication on public perceptions during public health crises. The study findings indicate that during this public health crisis, YouTube Chinese media, whose audience mainly consists of overseas Chinese-speaking users, primarily incorporated the enhancing strategies, succeeded by the diminish strategies, with limited application of deny strategies, while the use of rebuild strategies was virtually absent in this context. In addition, the research analysis confirms that Chinese media effectively increased the public’s positive perceptions of crisis events through crisis communication. Particularly, enhancing strategies proved most effective in improving public perceptions, followed by diminish strategies. In contrast, deny strategies failed to influence public perceptions of the crisis, and rebuild strategies demonstrated a negative impact on public perception. Thus, the research findings of this paper extend essential insights for effectively managing potential public health crises in the future.

## 1. Introduction

Effective crisis communication is crucial for successful crisis management. It helps in garnering favorable constituent attribution and increasing positive perceptions of stakeholders regarding the crisis event, thereby minimizing potential reputational harm [1]. In recent years, public health crises have severely affected the reputations of involved nations [2]. If not addressed, this negative impact could further hinder a country’s ability to appeal to tourists, attract foreign investments, and promote the export of goods and services [3,4,5]. Therefore, formulating effective crisis communication strategies (CCS) to mitigate the impact of such crises on national reputation has become a shared concern for both policymakers and research scholars.

Existing literature on crisis communication strategies is based on attribution theory, which holds that individuals are motivated to search for the potential causes of unexpected and adverse events, and these attributions of responsibility invoke negative reactions and sentiments [6]. Therefore, to mitigate the potential negative impact on an organization’s reputation from such attributions, crises such as environmental disasters, managerial corruption, product recalls, financial failures, and information leaks, all require effective and ethical communication with internal and external stakeholders [7]. Building on the attribution theory, Coombs (1995) proposed the Situational Crisis Communication Theory (SCCT) to analyze the patterns of crisis response. Since then, the SCCT emerged as a dominant approach in CCS [8].

Owing to the ascent of social media in recent years, social media platforms have evolved into essential channels for organizations to communicate with their stakeholders during crises [9,10,11,12]. During crisis events, enterprises can adeptly engage a large number of diverse stakeholders through social media, which significantly enhances the effectiveness of crisis communication strategies [13]. Pertinent studies have put forward certain good practices for the utilization of social media in crisis communication [14]. For instance, the Organization for Economic Co-operation and Development (OECD) advocates harnessing social media in order to raise public awareness of possible risks and crises and ensure monitoring and situational awareness while identifying survivors and victims of the crises [15]. Similarly, the Red Cross posits that efficacious crisis communication through social media entails an appropriate and peaceful tone, persistent monitoring, and a realization that traditional media is not replaced by social media [16].

Though there is a consistent interest from different disciplines, studies on crisis communication demonstrate certain limitations. First, most existing studies predominantly focus on the crisis communication of organizations or corporations; there is limited research on crises that could potentially affect a nation’s overall reputation. In fact, a significant national crisis would engage more stakeholders and have lasting adverse consequences for socio-economic growth. Second, the latest studies on crisis communication through social media largely center on blog-type platforms such as Twitter and lack empirical evidence from video-sharing platforms [17,18]. Apparently, validating the effectiveness of crisis communication strategies on such platforms is essential, given the prominence of platforms such as YouTube and TikTok. Lastly, data in empirical studies frequently face challenges associated with accuracy and effectiveness. For instance, non-random sampling methods and small sample sizes can lead to survivor bias, selection bias, or other methodological drawbacks.

In order to bridge the research gap discussed above, this study aims to evaluate the effectiveness of crisis communication strategies during significant national crises. Specifically, to accomplish this objective, this study analyzes the crisis communication strategies adopted by Chinese media on YouTube during the COVID-19 pandemic and assesses their impact on the perceptions of the overseas Chinese-speaking audience, a key stakeholder group for China. Additionally, this study explores the heterogeneity in crisis communication outcomes across different types of crisis communication strategies. The study findings suggest that during the COVID-19 pandemic, the crisis communication strategies employed by Chinese media on YouTube significantly increased the public’s positive sentiments. Among them, enhancing strategies were applied most frequently and proved to be the most effective.

This research article extends several valuable contributions to the present literature. First, from a national perspective, this study conducts an empirical analysis of crisis communication during a public health emergency, providing a valuable addition to empirical research on crisis communication in the face of significant national crises. Second, this study uses the sentiment index of stakeholders as a key metric for communication outcomes to evaluate the effectiveness of the four groups of crisis communication strategies proposed by SCCT on video-sharing platforms, thereby enhancing the empirical research of this theory in the internet era. Third, various econometric methods are integrated into this study to analyze big data from YouTube. Consequently, the application of econometric techniques and big data not only facilitates more diverse analyses but also enhances the robustness of empirical findings.

## 2. Research Background and Relevant Literature

This section offers an overview of the historical progression of the crisis event under investigation in this paper, followed by an introduction to the relevant literature.

### 2.1. The COVID-19 Pandemic

In January 2020, the COVID-19 epidemic spread rapidly in China, exerting a profound and lasting impact on the society and global economy [19,20]. This subsection provides an overview of the evolution of China’s COVID-19 pandemic by drawing data from the white paper “Fighting COVID-19: China in Action” issued by the State Council Information Office of China [21].

Primarily, the initial phase of China’s COVID-19 epidemic extended from 27 December 2019 to 19 January 2020. On 27 December 2019, the Hubei Provincial Hospital of Integrated Chinese & Western Medicine reported several cases of unexplained pneumonia to the Wuhan Jianghan Center for Disease Control and Prevention (CDC). Afterward, on 31 December 2019, the website of the Wuhan City Health Commission (WCHC) released the information circular related to pneumonia cases in Wuhan in order to confirm 27 active cases of pneumonia. Subsequently, the public was advised to avoid crowded venues, social gatherings, and poorly ventilated enclosed spaces. Thereafter, on 12 January 2020, the WCHC changed “viral pneumonia of unknown cause” to “pneumonia caused by the novel coronavirus” in information on its website. At the same time, the CDC China shared the genome sequence of the novel coronavirus with the World Health Organization (WHO). Subsequent to this, the proposed sequence was globally disseminated by the Global Initiative on Sharing All Influenza Data.

Subsequently, the second phase of the COVID-19 pandemic in China spanned from 20 January 2020 to 20 February 2020. On 20 January 2020, the academician Zhong Nanshan (head of the National Health Commission’s expert group) discussed the prevention and control measures of COVID-19 in a China Central Television (CCTV) interview as a response to COVID-19 concerns. This constituted the first official report by China’s “mainstream media” on the COVID-19 outbreak, thus signifying the commencement of strategic crisis communication. On 23 January 2020, Wuhan temporarily closed all outbound routes from the city’s railway stations and airports. Then, on 19 February 2020, China achieved initial progress in containing the virus when the number of newly cured and discharged cases surpassed the number of newly confirmed cases in Wuhan for the first time.

The third and fourth phases of China’s COVID-19 epidemic spanned from 21 February, 2020 to 28 April 2020. During this time period, the rapid transmission of the COVID-19 virus was effectively curbed in both Wuhan and the broader Hubei Province. Specifically, the daily count of new cases consistently remained in single digits since mid-March 2020. Subsequently, the outbound traffic restrictions for Wuhan and Hubei Province were removed on 8 April 2020. Eventually, the final hospitalized COVID-19 patient in Wuhan was discharged on 26 April 2020. This confirms the initial success of China in curtailing the COVID-19 spread within its mainland. The fifth phase of the COVID-19 pandemic commenced on 29 April 2020 in China, consequently reflecting a shift towards the normalization of control and preventive measures across the country.

### 2.2. Relevant Literature

#### 2.2.1. Situational Crisis Communication Theory (SCCT)

Coombs’ Situational Crisis Communication Theory (SCCT) identifies response strategies that organizations can use to handle a crisis. Evidently, when crises arise, different stakeholders involved in the crisis make attributions regarding the crisis responsibility. Thereafter, the adopted communication strategy of organizations hinges on the attributed responsibility levels (low, high, and minimum) and specific crisis types (accident, victim, and preventable crises) [22]. At the early stage, Coombs (1995) developed a decision tree for crisis managers, offering crisis response strategies based on crisis types [8]. Subsequently, Coombs further refined the SCCT [22,23]. As an illustration, Coombs (2007) enlisted a series of CCS from the lowest to the most accommodative level, including attacking the accuser, denial, scapegoat, excuse, justification, reminder, ingratiation, compensation, and apology [22]. Additionally, SCCT further extends a sound situational framework to ascertain when to employ certain CCS.

SCCT framework proposes four types of response strategies: deny, diminish, rebuild, and bolstering [8,22]. As shown in Appendix A Table A1, the deny strategies encompass three sub-strategies, namely, attacking the accuser, denial, and scapegoating. The diminish strategies consist of excuses and justification. Similarly, the rebuild strategies include compensation and apology. Finally, the bolstering strategies involve reminder, ingratiation, and victimage [22]. Moreover, as the practical application cases of SCCT have become more abundant, researchers have re-examined and extended SCCT’s response options [9,24,25,26]. For instance, some studies have introduced “enhancing strategies” [9,24]. Distinct from the bolstering strategy in SCCT, the enhancing strategy is referred to as introducing stakeholders to the “current” good works of the organization instead of the past good works [9,24].

#### 2.2.2. Evaluation of Crisis Communication Strategies

Based on empirical evidence, many scholars have assessed the effectiveness of crisis communication strategies in various crisis situations and have proposed numerous insights to improve the outcomes of crisis communication. For example, a study on the public health crisis suggests that the rebuild strategies were the predominant approach in most hotels’ response to the COVID-19 crisis, which successfully moderated the effects of affective evaluation and cognitive effort on customer satisfaction [27]. Similarly, a study on food safety issues revealed that the public with different characteristics has varied reactions to the same crisis communication strategy; notably, most stakeholders reported positive responses to rebuild strategies [28]. Besides, in victim crises, stakeholders perceive the organization as less responsible for the events, and they are also more inclined to empathize with the organization. Therefore, employing an apology strategy proves more effective for reputation repair than a denial strategy in such situations [29]. Conversely, in preventable crises, deny strategies are, nonetheless, effective for restoring corporate image when compared with diminish or acknowledge/rebuild strategies [30].

Additionally, researchers have shown particular interest in “silence” as a crisis communication strategy. A study categorizes the silence strategy into three types: delaying, avoiding, and hiding silences. Among these, avoiding and hiding silences intensified crises and negatively impacted the organizational image once forcefully broken, while delaying silence helped maintain or restore the image with primary stakeholders if it was successfully upheld and then broken as planned [31]. Conversely, an alternate perspective asserts that “silence” is not a viable strategy as it indicates the organization lacks control [32]. Furthermore, certain studies have challenged the SCCT by presenting alternative suggestions. For instance, an experiment-based study found that adopting rebuild strategies in a preventable crisis may lead to a backlash [30]. Similarly, another study further argued that rebuild strategies may not be consistently effective in preventable crises, as admission of responsibility can be perceived as an admission of organizational ineptitude, potentially inciting greater consumer resistance [33]. Moreover, a field study of collegiate participants refutes the idea that different crisis communication strategies have heterogeneous effects, suggesting that stakeholders’ personality traits determine reputation repair outcomes regardless of the communication strategy used [34].

Although the effectiveness of crisis communication strategies has attracted the attention of scholars, there is limited research concerning crisis communication in non-Western cultural contexts. The unique characteristics of Chinese social media determine that crisis communication between Chinese organizations and their stakeholders has its own distinct features [35]. In addition, crisis communication between Chinese organizations and their overseas stakeholders might present an entirely different scenario. To address the aforementioned research gap, this paper analyzes crisis communication strategies employed by Chinese media on YouTube during the COVID-19 outbreak and evaluates the effectiveness of different strategies. This research not only complements empirical research of SCCT but also unveils the distinct characteristics of Chinese media’s crisis communication strategies abroad.

## 3. Data Processing and Research Methodology

This section outlines the process for acquiring data from public sources and the subsequent data processing steps and introduces the research methodology used in this study.

### 3.1. Theoretical Concepts

Three concepts that run throughout this paper require further introduction: media, their roles in crisis communication, and the public.

Media. In this study, “Media” refers to online media or video bloggers who disseminate news on video-sharing platforms. The professional quality criteria of media content is captured by its number of followers and years of operation; other characteristics of media content are measured by observable variables such as location (Mainland China, Hong Kong, Taiwan, or other regions), type (independent media or institutional media), and language (Traditional or Simplified Chinese), among others.The role of media in crisis communication. In the subsequent data collection and processing section, this study further narrows the scope of the media samples. Firstly, the study limits the samples to news and politics media, whose function is to disseminate and analyze news, thus shaping public perceptions. Secondly, the study restricts media samples to “Chinese media”, referring to Chinese channels/vloggers and pro-China channels/vloggers that primarily use the Chinese language on YouTube. While these constraints don’t guarantee that the media’s objectives during the COVID-19 pandemic were centered on crisis communication (although intuitively they might have been), they do ensure that the content created by these media contains a significant amount of crisis communication information, as shown by text analysis of their video titles. In other words, while the media may not have explicitly intended to engage in crisis communication, the videos they released, which may contain crisis communication content, effectively played a significant role in crisis communication. Therefore, by analyzing videos with crisis communication content and contrasting them with others, the effectiveness of specific crisis communication strategies can be assessed.The public. In this study, the public primarily includes the active Chinese-speaking audience on YouTube, as they represent the most closely related stakeholders overseas for China and are the primary target group for crisis communication. YouTube, as the world’s largest video-sharing platform, hosts a substantial body of video channels/bloggers sharing different types of content. This platform attracts a large audience who view videos and express personal opinions. As of January 2022, YouTube had 2.562 billion monthly active users worldwide [36]. Therefore, the target audience for crisis communication is not necessarily local, especially considering YouTube’s limited reach in mainland China.

Reportedly, the proportion of YouTube users among the population aged 18 and above is notably high in Taiwan, Hong Kong, and countries/regions with an overseas Chinese population of over one million. Specifically, the proportion of YouTube users is nearly 90% in locations such as Australia, Canada, Singapore, Hong Kong, and Taiwan [37]. Consequently, the public primarily comprises the active Chinese-speaking audience from these countries and regions. Furthermore, researchers assume that YouTube Chinese-speaking channels/vloggers are a central platform linking China with its most closely related stakeholders overseas, thus serving as a crucial medium for crisis communication. As a result, the micro-level data from this platform are strongly representative.

### 3.2. Data Collection

In this study, micro-level data was acquired through the YouTube API, which is publicly available, and for the analysis, we used only publicly available data. The channels/vloggers from which we collected data are public YouTube entities. User content from such entities is also publicly accessible unless restricted by their privacy settings, in which case it is not included in the dataset. Our collection and analysis method comply with the terms, conditions, and policies of YouTube. First, random channels/vloggers were selected to achieve a representative sample. Subsequently, all the videos that were accessible to the public within the selected channels/vloggers were collected by the researchers. Finally, all comments were gathered from the sampled videos. In total, this study collected 14,645,269 audience comments from 173,216 YouTube videos for a time period between 31 December 2019 and 30 June 2020. The dataset encompasses detailed information about channels/vloggers, including their number of subscribers, language preference (simplified or traditional Chinese), and duration of media operation. Additionally, the dataset also provides metrics for individual videos, detailing titles, view counts, like counts, comment count, release dates, and the content of viewer comments. During the process of data collection and processing, the following principles were observed:The approach for building the sampling frame of YouTube video channels/bloggers has been established by repeatedly searching for top keywords from top news in Mainland China and subsequently extracting video channel/blogger information from the search results. First, the top news keywords were extracted from major events in Mainland China for the time period spanning from 2019 to 2020 [38,39]. Correspondingly, examples of these keywords can be found in Appendix A Table A2. Second, video information corresponding to each keyword was retrieved on a daily basis from 31 December 2019, to 30 June 2020 through the application programming interface offered by YouTube API Services. Video channel/blogger information was obtained from these results. Since the YouTube API services yield results for up to 50 videos per keyword search, each keyword search for a specific day was iterated until no new video data emerged. By utilizing the aforementioned sampling method, the study sample includes the vast majority of Chinese-speaking YouTube video channels/bloggers who have posted videos associated with Mainland China between 2019 and 2020, even when not all Chinese-speaking vloggers are included in the sample. This ensures that the study sample is as close as possible to the entire population or at least meets the criteria of random sampling for Chinese-speaking video channels/bloggers on YouTube.The research sample is further narrowed down to channels/vloggers with a higher level of relevance to Mainland China. Specifically, through text analysis of video titles, video channels/bloggers who had 10% or more of their videos related to Mainland China topics were selected. Clearly, the study scope targets channels/vloggers closely linked to Mainland China topics due to the study’s focus on Chinese media’s crisis communication on YouTube during the COVID-19 pandemic. The stated strategy is meant to maximize the inclusion of potential crisis communication channels of China and its stakeholders overseas in this crisis event while minimizing the complexities in sample expansion. In the robustness analysis section, various samples were generated based on varying criteria for judgment.In alignment with the research objectives, channels/vloggers in the “News and Politics” category were selected. YouTube classifies video content into 15 primary categories, such as “News and Politics”, “Sports”, and “Autos and Vehicles”, among others. Notably, channels/vloggers in the “News and Politics” category served as the primary crisis communication channels for China on YouTube during the COVID-19 epidemic. Analyzing the videos they disseminated offers direct insight into the crisis communication strategies of Chinese media and their effectiveness.Channels/vloggers with fewer than 10,000 subscribers were excluded from the study sample. In fact, the content distribution mechanism of YouTube exhibits strong selectivity when boosting videos from smaller channels. As a consequence, the inclusion of smaller channels/vloggers could result in non-random sampling. The less influential channels/vloggers are filtered out by setting a minimum subscriber threshold, thereby mitigating potential non-random sampling concerns. In addition, the robustness analysis section presents the regression outcomes for samples with a minimum subscriber count of 20,000 and 50,000, respectively.Channels/vloggers with a discernible pro-China ideological stance were selected from the pool of candidate channels/vloggers. The ideological tendencies of these channels/vloggers were manually classified by human coders based on the ideologies conveyed in their content. Specifically, channels/vloggers that predominantly reported on China in a positive/neutral manner were categorized as either Chinese media or pro-China media. Throughout the COVID-19 pandemic, these channels/vloggers served as the main channels for China’s crisis communication on YouTube, forming the focal point of this research. Consequently, this research finalized the study sample of Chinese media.Video samples are derived from the study sample of Chinese media. In fact, the initial video on COVID-19 from YouTube Chinese media appeared on 31 December 2019. Following the lifting of the lockdown in Wuhan on April 8th, videos on this topic began to diminish. Therefore, in accordance with the progression of the COVID-19 epidemic, the authors selected videos that were published from 31 December 2019, to 31 May 2020, as video samples for this study. Additionally, since this study primarily focuses on crisis management strategies of Chinese media, the authors further narrowed down the extracted video samples to those pertinent to Mainland China topics in order to satisfy the parallel trend assumption integral to the DiD method, which is employed to evaluate the effectiveness of crisis communication strategies. Furthermore, in the robustness analysis section, the sample period is extended to 30 June 2020, while incorporating videos irrespective of their relevance to Mainland China topics.Finally, audience comment samples are obtained from the video dataset. In particular, all audience comments are gathered from the video dataset, with comments in “Simplified Chinese” or “Traditional Chinese” being filtered as the audience comment samples. Owing to standardization challenges in performing sentiment analysis across diverse languages or Emojis, the researchers excluded comments in non-Chinese languages and those containing only Emojis. As a result, a mere 3.23% of the comments were excluded, exerting negligible influence on the study conclusions due to their limited information.

Eventually, the final sample of this study comprises 104 video channels/vloggers, 13,253 YouTube videos, and 1,790,816 audience comments. Information regarding the data collection process and the basic structure of the data can be referenced in Appendix A Figure A1.

### 3.3. Data Processing

#### 3.3.1. Text Analysis

In this paper, video content is identified through video titles. First, in order to filter out videos on Mainland China topics and those relevant to the COVID-19 epidemic, the following corpora were established: “Mainland China Corpus” and “COVID-19 Epidemic Corpus”. Second, using the Dalian University of Technology Sentiment Lexicon [40], corpora were created to classify different crisis communication strategies: “Denial”, “Diminish”, “Rebuild”, and “Enhancing Strategies Corpus”. In fact, these corpora were developed using a combination of manual vocabulary selection and word frequency analysis, and they were iteratively refined for accurate classification.

Subsequently, the “jieba” Chinese text segmentation module is utilized to perform word segmentation on video titles [41]. Thereafter, the segmented words were matched with the predefined corpora to identify videos on Mainland China topics, those associated with the COVID-19 epidemic, and to ascertain the crisis communication strategies employed. For instance, the video title “Salute to the medical staff in Wuhan fighting on the front line against the COVID-19 pandemic” is segmented into words such as “Salute”, “to”, “the”, “medical staff”, “in”, “Wuhan”, “fight”, “against”, “the”, “COVID-19”, “pandemic”, etc., which were thereafter matched with terms from the predefined corpora. Terms such as “COVID-19” and “Wuhan” matched with entries in the “COVID-19 Pandemic Corpus” and “Mainland China Corpus”, respectively, while the term “Salute” matched with an entry in the “Enhancing Strategy Corpus”. Consequently, this video is classified as relevant to topics in Mainland China and the COVID-19 pandemic and is labeled as adopting an enhancing strategy. It’s evident that a single video might be labeled as using multiple crisis communication strategies based on the content of the video title. This is consistent with the real-world scenario where crisis communication strategies are often applied in combination.

Additionally, considering the limitations of the “jieba” segmentation library, which may pose challenges in handling certain “new” vocabulary, the “jieba” segmentation corpus is enhanced by incorporating vocabulary from sources such as “China Scenic Spots Thesaurus” and “Chinese Thesaurus of Geographic Names” offered by Sogou, thereby improving its accuracy. Through testing within the scope of the study samples, the text processing method delivered an accuracy rate of 93.5 percent in identifying videos relevant to Mainland China, 97.6 percent in selecting topics related to the COVID-19 pandemic, and 82.6 percent in categorizing crisis communication strategies.

#### 3.3.2. Sentiment Analysis

In this study, sentiment analysis is applied to examine shifts in public perceptions of crisis events during the COVID-19 epidemic. Sentiment analysis is the process of analyzing digital text to determine if the emotional tone of the text is positive, negative, or neutral [42]. In line with this, the sentiment analysis on YouTube audience comments allows the researchers to depict changes in public perceptions of crisis events amidst the crisis communication process, thus enabling a more in-depth analysis of the effectiveness of crisis communication strategies [43]. In this study, the Dalian University of Technology Sentiment Lexicon is utilized as the foundational database for carrying out sentiment analysis on audience comments. Accordingly, the computational model is expressed in Equation (1) as follows [44].
(1)$$SI=\left(\frac{\sum_{i=1}^{n}V_{i}\cdot P_{i}\cdot N_{i}\cdot D_{i}}{\sum_{i=1}^{n}\left|V_{i}\cdot P_{i}\cdot N_{i}\cdot D_{i}\right|}+1\right)/2$$
where SI stands for the sentiment index of audience comments, ranging from 0 to 1. In fact, an estimated SI value greater than 0.5 implies a positive sentiment, nearing 1 for strong positivity. Conversely, an SI value less than 0.5 denotes a predominantly negative sentiment, with values approaching 0 confirming strong negativity. Besides, an SI value of 0.5 connotes a neutral sentiment, consequently suggesting that the audience’s comment is emotionally unbiased. Subsequently, the subscript i represents emotional words within the segmented text. Moreover, V_i_ means the emotional intensity of the word i, with measurement criteria sourced from the Dalian University of Technology Sentiment Lexicon [40]. Further, the lexicon was refined in order to align with the textual characteristics/attributes of YouTube audience comments. Parallel to this, P_i_ presents the polarity of the emotional word i, with 1 standing for positive words, −1 representing negative words, and 0 denoting neutral expressions. Similarly, N_i_ indicates the use of negative modifiers for the emotional word i, thereby assigning 1 for even counts of negative terms and −1 for odd instances. Meanwhile, D_i_ shows the usage of degree adverbs modifying the emotional word i, with greater modification corresponding to larger values [45].

Based on the computed SI values, this research paper derived two dependent/explained variables, namely: (1) sentiment index for audience comments and (2) a dummy variable representing positive sentiment. Besides this, the accuracy of the sentiment analysis method is assessed within the study sample. Evidently, the automated sentiment classifications (neutral, positive, or negative) in 72.56% of cases aligned with the manual identifications by researchers. Owing to the unconventional grammar and complex semantics of Chinese YouTube comments, the results of the model exhibit a relatively high accuracy [46].

### 3.4. Research Methodology

Despite the absence of evidence linking the origin of the COVID-19 virus to China, as the first country to experience the pandemic, the public is likely to attribute the losses triggered by the pandemic to China. These attributions are expected to stimulate negative sentiments and reactions towards China, thereby adversely impacting its global reputation [6,22]. Therefore, during the COVID-19 pandemic, it is essential for Chinese media to leverage social media platforms for effective crisis communication, aiming to reduce public animosity and mitigate potential reputational damage. Based on the SCCT, this study analyzes the relationship between crisis communication strategies and public perception through the following approach.

First, by conducting a text analysis on video titles, four tags corresponding to the expanded SCCT crisis communication strategies are assigned to pandemic-related videos. These tags specifically represent videos containing strategies of deny, diminish, rebuild, and enhancing. It should be noted that the enhancing strategy is not part of Coombs’ original theory; it was introduced by Kim & Liu (2012) [24]. In contrast to the bolstering strategy in SCCT, the enhancing strategy focuses on acquainting stakeholders with the organization’s “current” good works rather than its past achievements. Kim & Liu (2012) note that government organizations often employ the enhancing strategy to highlight their current initiatives [24]. This alignment with the cases presented herein leads the study to substitute Coombs’ (2007) bolstering strategy with the enhancing strategy [22].

Subsequently, the development process of China’s COVID-19 pandemic is classified into two stages in the context of crisis communication, including the initial stage of crisis communication (Stage I) and the strategic crisis communication stage (Stage II). During the initial outbreak, the COVID-19 virus exhibited rapid spread. Notably, comprehensive news coverage from China’s “mainstream media” remained absent. This suggests the crisis communication corresponding to this public health crisis was predominantly in its initial stage, termed Stage I in our study. On 20 January 2020, CCTV first published specialized reports pertinent to the COVID-19 pandemic, hence marking the launch of strategic crisis communication. Consequently, the period following January 20, 2020, is identified as the strategic crisis communication stage (Stage II).

Finally, the efficacy of crisis communication strategies is evaluated using the difference-in-differences (DiD) method. In this paper, the difference in public perceptions of crisis events between Stage I and Stage II constitutes the first difference in the DiD framework. In Stage I, strategic crisis communication was not initiated, whereas COVID-19 pandemic-relevant videos presented fragmented crisis communication information. Therefore, public perceptions of crisis events should not be expected to be impacted by crisis communication strategies during this stage. In Stage II, with the initiation of strategic crisis communication, the strategies employed are likely to significantly influence public perceptions of the crisis. Additionally, the second difference in the DiD framework is associated with differential sentiments expressed by the audience in different thematic video contents. Certainly, videos employing specific crisis communication strategies (the treated group) are expected to influence public perceptions, which will be evident in the shift in public sentiment. Conversely, videos with varied themes (the control group) should not have a consistent impact on public sentiment. The DiD approach, considering the two differences mentioned, can estimate the influence of crisis communication strategies on public perceptions of crisis events.

Specifically, the commencement of strategic crisis communication on 20 January 2020, is perceived as a policy shock, with Stage II representing the policy implementation window. Using the DiD approach, the researchers estimate the changes in public perceptions of crisis events caused by this policy shock in order to discern the efficacy of crisis communication strategies. Accordingly, the baseline regression model is expressed in Equation (2) as follows:(2)$$Y_{ijkdl}=\beta_{0}+\beta_{1}\cdot {StageⅡ}_{j}+\beta_{2}\cdot {CCS}_{jkdl}+\beta_{3}\cdot \left({CCS}_{jkdl}\cdot {StageⅡ}_{j}\right)+\beta_{4}\cdot X_{ijkdl}^{\prime }+u_{k}+u_{d}+u_{l}+\varepsilon_{ijkdl}$$
where the explained/dependent variable Y denotes the measure of public perceptions. Referring to the previous literature, “audience sentiment” is adopted to estimate public perceptions of crisis events as a critical outcome of crisis communications [47,48]. Specifically, sentiment analysis is performed on viewer comments in order to derive (1) a sentiment index and (2) a dummy variable representing positive sentiment. Apparently, in videos focused on crisis communication, these variables capture public perceptions of crisis events. Conversely, in videos unrelated to specific crises, the variables indicate viewers’ emotional reactions. Furthermore, subscripts i, j, and k stand for video comments, videos, and video channels/bloggers from YouTube, respectively. Similarly, subscript d connotes the publication date of the video, while l signifies the video topic.

In addition, “Stage II” serves as a dummy variable, with this variable set to 1 when video j is published post-January 20, and 0 otherwise. Besides, “CCS” is a dummy variable that distinguishes between the treatment and control groups. In the baseline regression, when “CCS” equals 1, it explicitly denotes that videos contain one or more crisis communication strategies. To satisfy the parallel trends assumption, “CCS” set to 0, indicates that videos are unrelated to crisis communications but related to Mainland China topics. Moreover, X’ represents control variables, which contain information such as the number of subscribers for the video channels/bloggers’, video channels/blogger’s location, language (Simplified Chinese/Traditional Chinese), and years of operation. Furthermore, u_k_ implies fixed effects for video channels/bloggers, thereby controlling for all channel/blogger-level characteristics that do not vary over time. Thereafter, u_d_ and u_l_ capture fixed effects for publication time and themes of videos, respectively. As the fixed effects of video publication timing and themes are controlled, the coefficients of dummy variables “Stage II” and “CCS” shall be absent in the regression results.

Finally, β_3_ signifies the estimated coefficient of interest, reflecting the average treatment effect on the treated (ATT). This estimated coefficient illustrates the influence of crisis communication on the public perceptions of crisis events, which is expected to be positive. Noticeably, under the condition of eliminating interference from other influencing factors, if there is a significant increase in the likelihood of observing positive sentiment or the audience sentiment index rises in videos that contain crisis communication strategies, this indicates that viewers express more positive sentiments towards the crisis events. This suggests that the crisis communications strategy has achieved positive outcomes.

### 3.5. Summary Statistics

The Chinese media played an active role in crisis communication during the COVID-19 pandemic, leading to changes in public perceptions of this crisis. Accordingly, Table 1 highlights the differences in public sentiment between Stage I and Stage II. As measured by the audience sentiment index or the proportion of positive sentiment, the public demonstrated a lower degree of positive sentiments during the initial stage of crisis communication (Stage I). Subsequently, public sentiment significantly improved after implementing strategic crisis communication in Stage II. Prominently, in terms of the positive sentiment proportion, the mean difference between videos containing CCS (Treatment group) and those on other topics (Control group) was −0.12 in Stage I, whereas the mean difference stood at −0.01 in Stage II. Additionally, the statistical analysis reflects that the mean sentiment index difference between the treatment and control groups was more obvious in Stage I than in Stage II. This provides initial evidence supporting the postulated hypothesis: effective crisis communication improves public perceptions associated with crisis events. Moreover, Appendix A Table A3 extends more basic information on other parameters within the study sample.

## 4. Research Results

Primarily, this section examines the CCS employed by Chinese media during the outbreak of the COVID-19 pandemic. Subsequently, the impact of strategic crisis communication on public perceptions of crisis events is investigated. Afterward, parallel trend tests are carried out by the researchers. Thereafter, a robustness analysis is performed in this section. Finally, this study explores the heterogeneous effect of different crisis communication strategies.

### 4.1. CCS Adopted by Chinese Media during the COVID-19 Pandemic

Table 2 presents a phased analysis of the crisis communication strategies implemented. In the initial stage of crisis communication (Stage I), due to limited information from mainstream sources, Chinese media did not present a unified crisis communication strategy. Therefore, at this stage, Chinese media’s coverage of the COVID-19 epidemic adopted relatively diversified crisis communication strategies, including enhancing, diminish, and deny strategies.

On 20 January, CCTV broadcasted its first special report on the COVID-19 pandemic, thereby initiating strategic crisis communication. Subsequently, Chinese media increased their coverage of the crisis, and their crisis communication strategies began to show a unified approach. Relevant data reveal that Chinese media predominantly adopted the enhancing strategies during the second phase. While employing the reminder sub-strategies, Chinese media informed the public about the prevention measures undertaken by various governmental bodies. Similarly, the ingratiation sub-strategies were used to highlight and praise the collective efforts across society to combat the COVID-19 pandemic.

During the third phase of the pandemic, Chinese media increasingly incorporated the enhancing strategies, with a simultaneous uptick in the application of the diminish strategies. In the fourth and fifth phases, Chinese media further intensified their use of the diminish strategies, with the domestic spread of COVID-19 under control. Meanwhile, the application of other strategies remained relatively consistent.

From 31 December 2019 to 31 May 2020, enhancing strategies were the most frequently utilized, evident in 85.22 percent of videos addressing the COVID-19 epidemic. Furthermore, diminish strategies were the next most utilized, employed in 26.62 percent of the videos. Similarly, deny strategies were applied in 13.32 percent of the cases, ranking third. It is worth noting that rebuild strategies saw minimal application during this period. Furthermore, deny strategies were most commonly utilized during the first stage of the COVID-19 pandemic’s development, reaching 23.91 percent. Additionally, the use of diminish strategies peaked during the fifth stage, comprising 37.64 percent. Meanwhile, enhancing strategies were most widely used during the fourth stage, with a significant 87.49 percent share.

### 4.2. The Impact of Strategic Crisis Communication on Public Perceptions

Table 3 presents the results of baseline regression estimation. The analysis covers the period from 31 December 2019 to 31 May 2020. The dependent/explained variables are (a) the dummy variable for positive comments and (b) the sentiment index. Control variables comprise information such as the number of subscribers for the video channels/bloggers, video channels/blogger location, language (Simplified Chinese/Traditional Chinese), dummy variables for video channels/bloggers, video themes, and video posting times. Furthermore, standard errors (SEs) are adjusted through video channel/blogger-level clustering. Moreover, the control variables and clustered standard errors (SEs) remain consistent in subsequent regression analyses. The benchmark regression outcomes conform to the study’s anticipations. As depicted in Table 3 (a), the crisis communication strategies employed by Chinese media exhibit considerable effectiveness. In videos where crisis communication strategies were employed, positive comments rose by 4.10 percent in Stage II. Besides, these influences are significantly different from zero (0) at the 1 percent level of statistical significance.

Table 3 (b) displays regression results using the sentiment index as the dependent variable. The derived results also illuminate the significant effectiveness of crisis communication. In videos where crisis communication strategies were incorporated, the sentiment index witnessed an average increment of 0.287 during Stage II. Given the study’s sentiment index model—where the most positive sentiment is valued at 1, and the most negative sentiment is valued at 0—the rise in audience sentiment is prominently significant.

### 4.3. Parallel Trends Test

The treated and control groups satisfying the parallel trends assumption serve as a prerequisite for the credibility of the aforementioned estimation outcomes. Therefore, an event study methodology is employed to ascertain the parallel trends assumption in order to ensure the reliability of the identification results. The specific model is expressed in Equation (3) as follows:(3)$$Y_{ijkdl}=\beta_{0}+\beta_{v}\sum_{v\geq -3}^{18}D_{d_{i0}+v}+\beta_{c}\cdot X_{ijkdl}^{\prime }+u_{k}+u_{d}+u_{l}+\varepsilon_{ijkdl}$$
where D_di0+v_ represents a set of dummy variables, with the subscript d_i0_+v indicating the week when a crisis communication-related video was uploaded to a channel on YouTube. Particularly, D_di0+v_ is set to 1 for the week v after the upload (i.e., when d − d_i0_ = v) and 0 otherwise. Additionally, January 20 is designated as the start of strategic crisis communication, marking the first day of week 0. The week preceding January 20 serves as the omitted reference group in Equation (3). Furthermore, the parameters group βv in the model captures the effect of CCS on public perceptions of crisis events, relative to the reference week.

Figure 1a,b illustrate the results of the parallel trends tests with the dummy variable for positive sentiment and the sentiment index as the explained/dependent variables, respectively. The coefficients of β_v_ are not significantly different from 0 during the second and third weeks preceding the crisis communication. This posits that prior to the onset of strategic crisis communication, there was no significant difference in the trend of public sentiment between the control group and the treatment group, hence validating the parallel trends assumption.

### 4.4. Robustness Test

In this section, the authors conduct multiple robustness tests to validate the reliability of baseline regression estimations.

First, researchers consider adopting different criteria to identify video channels/bloggers with a stronger association with Mainland China. In the previous approach, video channels/bloggers who had 10% or more of their videos related to Mainland China topics were selected. Currently, outcomes are generated when this relevance threshold is adjusted to 5% and 15%, respectively. The regression outcomes are reported in Table 4 (a,b). The significance levels and the magnitudes of the interaction term coefficients in these cases closely align with those in Table 3.

Second, different samples are created based on different subscription count thresholds. Previously, a sample of Chinese-speaking video channels/bloggers with a minimum subscription count of 10,000 was selected in this study. Currently, the paper examines samples with minimum subscription thresholds of 20,000 and 50,000, with the regression results illustrated in Table 4 (c,d), respectively. It is clear that all interaction term coefficients are statistically significant, and the directions and magnitudes of these coefficients are similar to those presented in Table 3.

Third, the sample period has been extended in this study. Table 4 (e) analyzes a video sample that spanning from 31 December 2019 to 30 June 2020, thus encompassing an extended duration of the fifth phase. Consequently, the regression outcomes closely reflect the baseline results, maintaining consistent conclusions.

Fourth, the scope of the sample is expanded to include video content unrelated to Mainland China. Initially, for the baseline regression, the study selected videos on Mainland China topics to satisfy the parallel trend assumption. Presently, a new regression analysis is conducted without such restriction. In line with this approach, Table 4 (f) presents regression results of all video samples from 31 December 2019 to 31 May 2020. Notably, the interaction term coefficients remain statistically significant, and the impacts on public perception of crisis events are consistent with the baseline regression estimations.

Fifth, given the binary nature of the dependent/explained variable, probit regression is applied for robustness tests to mitigate model specification bias. Specifically, the regression results are presented in Table 4 (g). The directions and significance levels of the coefficients are consistent with those in Table 3, which underscores the robustness of the baseline regression findings.

### 4.5. Heterogeneity Analysis

The empirical analysis in this paper confirms that the crisis communication of Chinese media positively impacted the public perceptions of crisis events during the COVID-19 pandemic. Intuitively, the efficacy of various crisis communication strategies is expected to vary. This section examines the heterogeneity among different crisis communication strategies.

Initially, four sub-samples were created, each corresponding to different crisis communication strategy used in the videos. Specifically, videos utilizing deny strategies were selected and paired with the control group to form sub-sample 1. Similarly, videos applying diminish strategies were isolated and merged with the control group to constitute sub-sample 2. Moreover, videos implementing rebuild strategies were extracted and integrated with the control group, creating sub-sample 3. Furthermore, videos with enhancing strategies were separated and combined with the control group to establish sub-sample 4. Subsequently, Equation (1) was estimated for each of the sub-samples 1–4 individually. As a result, regression results for each sub-sample illustrate the impact of the specific crisis communication strategies on public perception of crisis events.

Figure 2a illustrates the regression results where the dummy variable for positive sentiment serves as the explained/dependent variable. Of the four crisis communication strategies, the enhancing strategy significantly increased positive audience comments by 5.77%. The diminish strategy also succeeded, raising positive comments by 4%. These effects were statistically significant at the 1% and 5% levels, respectively. However, videos employing deny strategies did not show a significant increase in positive comments, indicating their limited effectiveness in crisis communication. Furthermore, it is worth noting that videos employing rebuild strategies significantly reduced positive audience comments by 3.42%, demonstrating a negative impact on public perception. Figure 2b presents the regression results with the sentiment index as the explained/dependent variable. It confirmed the effectiveness of the enhancing and diminish strategies. Similarly, the deny strategies showed no significant effect, and the rebuild strategies adversely affected public perception.

## 5. Discussion

The findings of this study suggest that the enhancing strategy was the primary approach adopted by Chinese media during the pandemic. By implementing the reminder sub-strategy, stakeholders became aware of the government’s efforts in pandemic prevention. Similarly, the authorities employed the ingratiation sub-strategy to highlight the collective efforts across society in curbing the spread of the epidemic. Furthermore, the victimage sub-strategy was implemented to underscore the challenges and sacrifices made by healthcare professionals during the epidemic. As the domestic spread of the pandemic gradually came under control, Chinese media increasingly implemented the diminish strategy, aiming to further assist in improving the public perceptions of the crisis. Notably, the use of the deny strategy was comparatively limited, and the rebuild strategy was seldom applied in crisis communication.

While this study focuses on the crisis communication of Chinese media on global social media platforms, the strategies they employ share notable similarities with those used within Mainland China. For instance, Coombs (2017) categorizes nine sub-strategies of crisis communication from least to most accommodative [22]; Chinese media predominantly employ mid-level strategies, including excuse, justification, reminder, and ingratiation. This highlights Chinese media’s crisis communication features: striving for the “golden mean” and avoiding extreme strategies, consistent with previous studies [35,49]. Moreover, the infrequent use of rebuild crisis response strategies by Chinese media corresponds with Cheng’s (2016) findings on the crisis communication strategies of the Red Cross Society of China [50].

Benchmark regression results confirm that crisis communication significantly improves public sentiment, highlighting its effectiveness in improving public perceptions of crises. This provides a valuable addition to the existing literature and aligns with our expectations. Furthermore, the heterogeneity analysis points out that enhancing and diminish strategies exhibited positive impacts out of the four crisis communication strategies delineated by SCCT, in contrast to the insignificant results of deny strategies and the negative outcomes of rebuild strategies in this specific crisis. These observations are consistent with some previous research results. For instance, previous literature suggests that rebuild strategies can elicit negative responses in preventable crises [30,33], consistent with this study’s observations. Nevertheless, they also contradict the findings of some other studies. A previous finding suggests that the apology strategies surpass deny strategies in repairing reputation during victim crises [20]; however, this study observes contrasting outcomes.

Various studies demonstrate that the effectiveness of crisis response strategies is primarily influenced by the crisis’s nature and the characteristics of the involved organization [46,51]. These studies also reveal that the efficacy of crisis communication strategies is dynamic and subject to continual change. In fact, Coombs (2007) highlights that crafting a single, definitive list of crisis response strategies is unrealistic for researchers [22]. Instead, as a rapidly evolving practice area of scholarly research, new findings are continually emerging from crisis communication research, offering valuable insights for practitioners in the field [23]. This study provides a new assessment of the effectiveness of SCCT in the context of the social media era, confirming Coombs’ (2014) view that denial strategies must be applied with caution. It suggests that any link, even a minimal one, between the organization and the crisis can exacerbate reputational damage if a denial strategy is employed [52,53]. Additionally, the findings align with Kim & Liu’s (2012) observation that governments often employ enhancement strategies to highlight their positive actions [24]. Consequently, replacing Coombs’ (2007) bolstering strategy with an enhancing strategy in SCCT is a practical and suitable modification, meeting the demands of contemporary contexts [22].

## 6. Conclusions and Policy Implications

This study confirms the positive influence of crisis communication on the public perception of crisis events and illustrates that adopting enhancing and diminish strategies during public health crises delivers improved outcomes. The proposed findings offer valuable implications for policymakers.

Firstly, the enhancing CCS is recommended for primary use in forthcoming public health crises based on the established effectiveness of the enhancing strategy in this incident. In the face of a public health crisis, a nation’s reputation is critically evaluated. At this moment, the implementation of the reminder sub-strategy highlights the proactive measures taken by the country. Parallel to this, employing the ingratiation sub-strategy encourages stakeholders to collaborate with the nation in crisis response. Collectively, these approaches help to increase stakeholders’ positive perceptions of crises while playing a central role in upholding the reputation of the nation.

Secondly, the diminish CCS also yielded effective outcomes during the COVID-19 crisis. Thus, the potential role of this strategy in public health crises should be adequately recognized. The diminish strategy reduces the proportion of the public attributing the event to the nation by either denying intent to inflict harm or minimizing the perceived damage stimulated by the crisis. Eventually, this alleviates public anguish and dissatisfaction while mitigating the adverse sentiments triggered by the event and concurrently plays a pivotal role in maintaining the nation’s reputation.

Finally, it is advised to be cautious and, where possible, avoid adopting rebuild strategies when handling significant national crises. Adopting rebuild strategies in a preventable crisis may provoke backlash, as it might be interpreted as conceding to governmental failures, thereby increasing the public’s negative perception of the government. Particularly during the COVID-19 pandemic, negative perceptions of governmental leadership may exacerbate the crisis, compromising the efficacy of subsequent crisis management measures and further leading to catastrophic consequences. Therefore, the rebuild strategies should not be considered a practical strategy for national emergency response.

## 7. Limitations and Future Direction

This study possesses inherent limitations. First, by analyzing the crisis communication strategies adopted by Chinese media on YouTube during the COVID-19 pandemic, the study aims to investigate the influence of strategic crisis communication on public perceptions during public health crises. However, the unique characteristics of Chinese media and China’s overseas stakeholders may limit the generalizability of these findings [35]. While the impact of Chinese media’s crisis communication on Chinese overseas stakeholders’ perceptions might not extend to other countries, particularly developed ones, it could be representative of emerging countries with situations akin to China’s.

Second, this paper employs a difference-in-differences (DiD) model to determine the causal relationships between crisis communication strategies and public perception. In the baseline regression, time dummy variables are controlled to mitigate time trend effects; video channel/blogger dummy variables are included to account for the characteristics of video channels/bloggers that do not change over time; key characteristics of video channels/bloggers, such as the number of followers, operational duration, location, type, and language, are controlled to isolate the influence of the professional quality criteria of media content on the regression outcomes; it is presumed that videos from the same channel/blogger are correlated, leading to the use of clustered standard errors at the channel/blogger level for analysis. Additionally, parallel trend tests are carried out to verify that there are no substantial differences between the treatment and control groups prior to the policy shock. Robustness analysis is conducted to enhance the robustness of the study. These measures ensure, to the greatest extent, that the regression results can accurately interpret the impact of crisis communication strategies on public perception. However, due to the limited information collected, the study may still be subject to omitted variable bias, leading to biased regression coefficients. This represents a typical risk in econometric analysis and constitutes an unavoidable limitation of this study.

Third, our data utilization is confined. Specifically, the empirical analysis of this study relied exclusively on text, neglecting potential insights from videos and images. Videos and images, intuitively, may contain richer information, suggesting avenues for further research.

## Figures and Tables

**Figure 1 behavsci-14-00091-f001:**
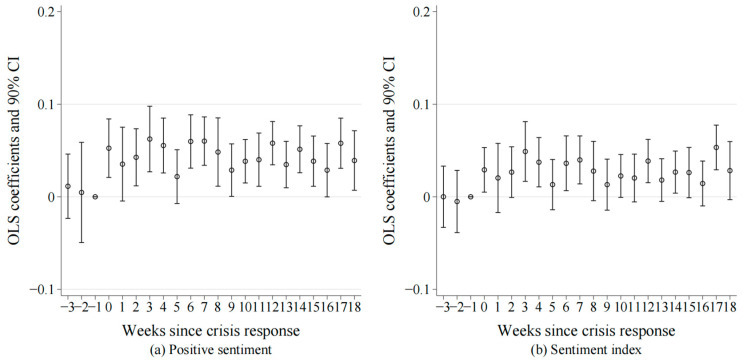
Parallel trend test. (**a**) Parallel trends tests with the dummy variable for positive sentiment as the explained/dependent variable. (**b**) Parallel trends tests with the sentiment index as the explained/dependent variable. The circle shapes represent the OLS. estimates of β_v_ at the horizontal axis where k equals, while the dashed line(s) depict the 90% confidence interval corresponding to those OLS estimates.

**Figure 2 behavsci-14-00091-f002:**
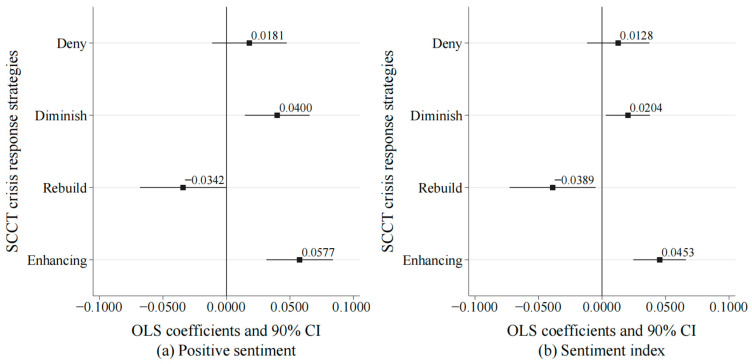
Heterogeneity analysis. (**a**) Regression results with the dummy variable for positive sentiment as the explained/dependent variable. (**b**) Regression results with the sentiment index as the explained/dependent variable. The square shapes represent the OLS estimates of corresponding sub-samples, while the solid line(s) depict the 90% confidence interval corresponding to those OLS estimates.

**Table 1 behavsci-14-00091-t001:** Summary Statistics.

	Mean	Observations	Mean	Observations	Mean Difference
Positive Sentiment	Videos containing CCS (Treatment group)		Videos unrelated to CCS (Control groups)		
Stage Ⅰ	0.28	1825	0.40	137,758	−0.12 ***
Stage II	0.38	524,247	0.39	1,126,986	−0.01 ***
Sentiment Index	Videos containing CCS (Treatment group)		Videos unrelated to CCS (Control groups)		Mean Difference
Stage Ⅰ	0.46	1825	0.55	137,758	−0.09 ***
Stage II	0.53	524,247	0.54	1,126,986	−0.01 ***

Note: *** denotes significance levels at the 1%. The time range of the sample is from 31 December 2019 to 31 May 2020. Unless otherwise specified, the same applies to the following tables and figures. CCS stands for crisis communication strategies.

**Table 2 behavsci-14-00091-t002:** Crisis Communication Strategy Applied During COVID-19 Pandemic.

		Deny	Diminish	Rebuild	Enhancing	Observations
Stage I	Stage 1	23.91	28.26	0	69.57	46
Stage II	Stage 2	14.57	18.65	0.33	82.89	1496
	Stage 3	11.94	23.75	0.70	86.56	863
	Stage 4	11.23	33.77	1.30	87.49	1158
	Stage 5	15.64	37.64	0.73	86	550
Total		13.32	26.62	0.73	85.22	4113

Note: The values in the table represent the percentage of videos that adopt the corresponding crisis communication strategies. A single video may contain multiple crisis communication strategies. Therefore, the sum of the percentages for different strategies might exceed 100%.

**Table 3 behavsci-14-00091-t003:** The Crisis Communication Outcomes of Chinese Media.

Variables	(a) Positive Sentiment	(b) Sentiment Index
CCS × Stage II	0.0410 ***	0.0287 ***
	(0.0142)	(0.0084)
Control variables	Yes	Yes
Fixed effect	Yes	Yes
Observations	1,790,816	1,790,816

Note: *** denotes significance levels at the 1%. The regression model incorporates video channel/blogger level clustered robust standard errors. Unless otherwise specified, the same applies to the following tables and figures.

**Table 4 behavsci-14-00091-t004:** Robustness Test.

Variables	(a) Positive Sentiment	(b) Sentiment Index
(a) Bloggers whose videos contained 5% or more of content related to mainland China		
CCS × Stage II	0.0413 ***	0.0291 ***
Observations	(0.0142)	(0.0084)
Observations	1,802,300	1,802,300
(b) Bloggers whose videos contained 15% or more of content related to mainland China		
CCS × Stage II	0.0374 **	0.0244 ***
	(0.0143)	(0.0079)
Observations	1,737,070	1,737,070
(c) Samples with subscription counts greater than or equal to 20,000		
CCS × Stage II	0.0412 ***	0.0288 ***
	(0.0142)	(0.0084)
Observations	1,783,276	1,783,276
(d) Samples with subscription counts greater than or equal to 50,000		
CCS × Stage II	0.0415 ***	0.0287 ***
	(0.0143)	(0.0085)
Observations	1,711,633	1,711,633
(e) Samples from 31 December 2019 to 30 June 2020		
CCS × Stage II	0.0409 ***	0.0295 ***
	(0.0133)	(0.0093)
Observations	2,235,291	2,235,291
(f) Samples (Including topics unrelated to mainland China) from 31 December 2019 to 31 May 2020		
CCS × Stage II	0.0493 ***	0.0366 ***
	(0.0157)	(0.0130)
Observations	3,613,356	3,613,356
(g) Probit regression		
CCS × Stage II	0.1178 ***	0.1095 ***
	(0.0402)	(0.0263)
Observations	1,790,816	1,790,816

Note: **, *** denotes significance levels at the 5%, and 1%, respectively.

## Data Availability

The data presented in this study are available on request from the corresponding author.

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
