# Peer review of "Influence of Strategic Crisis Communication on Public Perceptions during Public Health Crises: Insights from YouTube Chinese Media"

_behavsci, 2024, doi:10.3390/bs14020091_

Round 1

Reviewer 1 Report

Comments and Suggestions for Authors

Dear Authors,

the idea of your article is very interesting, however, it is necessary to supplement its theoretical basis.

The article has a clear structure, but its goals are repeated in almost every subsection.

In your article, you demonstrate a very good knowledge of how to analyze data from web platforms, a clear methodology and a well-constructed data sample. The data has been expertly analyzed using statistical analysis methods. The research results are credible and reliable, they are presented appropriately, showing interesting dynamics of content perception of the YouTube audience during the Covid pandemic.

The main problems are caused by two theoretical concepts used in the article: crisis communication and media. About the media. It is impossible to understand the findings of the article if there is no concrete definition of the media, if the analyzed sources are selected only according to the number of followers on YouTube, but the goals of their creation and the professional quality criteria of their content are not understood. Currently, data is collected from media, blogs, channels,  but there is a lack of information on how these sources are related to media functions and what their role is in crisis communication.

About crisis communication. Although you offer an "innovative" approach and claim that the media is engaged in crisis communication, there is no theoretical or empirical basis for this. Thus, the obtained data do not show either correlation or causality, conclusions can be drawn about the sentiment of active audience representatives, which can be influenced by many other factors, not only "crisis communication content" in various mediated sources.

Before you can submit an article for publication, you need to establish a clear conceptual framework.

I would recommend not to use what is mentioned in the conclusions about the impact of crisis communication on audience trust, because you have not studied trust nor the audience, the study contains data on the sentiment of active Youtube users (content commentators, interactively engaged with content or other users).

Best regards

Comments on the Quality of English Language

No comments.

Reviewer 2 Report

Comments and Suggestions for Authors

This is a high-quality paper.  The content described and contextualized with respect to previous and present theoretical background and empirical research. The literature review is appropriately structured, enough amount of sources cited relevant to the research, data processing and research methodology clearly stated. Narrative analysis is carried out properly: a program is written to read a huge array of texts (I must undeline positively also time slices), a corpus of the language of these texts is compiled, then compared with dictionaries of expressions, and based on this an index is compiled. Conclusions and policy implications provided.

Reviewer 3 Report

Comments and Suggestions for Authors

Thank you for the opportunity to familiarize with your research results. The idea to aims to evaluate the crisis communication strategies utilized by Chinese media on YouTube is relevant and brings added value to the field of crisis communication. I found the manuscript to be promising, however, it needs conceptual reframing. Please consider these remarks:

The authors referred to Coombs' Situational Crisis Communication Theory, but it's unclear why they reduced the number of strategies from eight to five. The rationale for this reduction should be clarified in both the theoretical justification and methodology sections.

The introduction of the term 'bolstering strategies' should be justified, as it's not part of Coombs' original theory. New concepts need to be properly justified when introduced.

Accurate citations are essential, and it's important to ensure that the authors' ideas remain intact when referencing other sources. Double-checking all citations to maintain the original context is recommended.

The design and purpose of the study should be clear, particularly with regard to the intended audience of the YouTube messages analysed. It is important to emphasise that YouTube communication may not be targeted at a local audience, especially given YouTube's limited reach in mainland China. The text states that "the proportion of YouTube users is nearly 90% in places such as Australia, Canada, Singapore, Hong Kong and Taiwan", but nowhere else does it emphasise that YouTube communication is not aimed at a local audience. Authors should clearly identify the audience they are studying (and why). Aligning the aim, methodology and conclusions/discussion with the intended audience is necessary to avoid misleading implications.

The discussion section, especially paragraph 3 (lines 671-681), needs to be rewritten for clarity, especially regarding contradictions and overlaps with previous studies.

Explaining the limitations of the study and critically assessing other limitations would improve the scope and depth of the study.

Addressing these issues will improve the clarity, coherence and overall scientific rigour of the manuscript.

Comments on the Quality of English Language

Only minor editing of English language required

Reviewer 4 Report

Comments and Suggestions for Authors

Thank you for giving me the opportunity to review the manuscript titled “Influence of strategic crisis communication on public perceptions during public health crises: Insights from YouTube Chinese Media”. I would like to congratulate the authors for providing a compelling justification for the study, well-explained methodology, and discussion sections.

The following are some minor comments.

Introduction

The research gap in the introduction is properly justified. However, the research’s novelty and problem statement can be improved and clarified.

Methodology

This section is properly explained. However, some insight about the limitations of the analysis techniques (benchmark regression, heterogeneity analysis etc.) will help to provide more understanding to the reader.

Implications

The theoretical implications with reference to Situational Crisis Communication Theory (SCCT) are missing.

Limitations and future directions

Make 'limitations and future direction" as separate heading

The manuscript highlighted the data utilization confinement as a limitation in the last paragraph of the discussion section. It is recommended to include other limitations related to the generalizability of results and statistical analysis.

Also, identify future areas of research in which the current investigation can be extended.

Round 2

Reviewer 3 Report

Comments and Suggestions for Authors

I thank the authors for all their corrections and for taking into account all my comments and suggestions. Nevertheless, the abstract and the introduction of the article should include a clear indication of the audience for the media content analysed in the study. In the rest of the article, the authors have provided this explanation in sufficient detail. 

Author Response

Dear Reviewer,

We sincerely appreciate your attentive review and valuable suggestions. In light of your insightful review comments, we have meticulously revised our manuscript, marking all changes in red for clear reference. The main revisions in the paper and our responses to your comments are as follows:

Point 1: I thank the authors for all their corrections and for taking into account all my comments and suggestions. Nevertheless, the abstract and the introduction of the article should include a clear indication of the audience for the media content analysed in the study. In the rest of the article, the authors have provided this explanation in sufficient detail.

Response 1: In both the abstract and introduction sections of the article, we explicitly identified the audience for the analyzed media content.

The reversed version is as follow:

Abstract: Crisis communication plays a crucial role in preserving the national reputation during significant national crises. From the perspective of Situational Crisis Communication Theory (SCCT), this research paper analyzed over 1,790,816 YouTube comments from Chinese-speaking audiences, using sentiment analysis alongside the Difference-in-Differences (DiD) model, in order to investigate the influence of strategic crisis communication on public perceptions during public health crises. The study findings indicate that during this public health crisis, YouTube Chinese media, whose audience mainly consists of overseas Chinese-speaking users, primarily incorporated the enhancing strategies, succeeded by the diminish strategies, with limited application of deny strategies, while the use of rebuild strategies was virtually absent in this context. In addition, the research analysis confirms that Chinese media effectively increased the public's positive perceptions of crisis events through crisis communication. Particularly, enhancing strategies proved most effective in improving public perceptions, followed by diminish strategies. In contrast, deny strategies failed to influence public perceptions on the crisis, and rebuild strategies demonstrated a negative impact on public perception. Thus, the research findings of this paper extend essential insights for effectively managing potential public health crises in the future.

In order to bridge the research gap discussed above, this study aims to evaluate the effectiveness of crisis communication strategies during significant national crises. Specifically, to accomplish this objective, this study analyzes the crisis communication strategies adopted by Chinese media on YouTube during the COVID-19 pandemic and assesses their impact on the perceptions of the overseas Chinese-speaking audience, a key stakeholder group for China. Additionally, this study explores the heterogeneity in crisis communication outcomes across different types of crisis communication strategies. The study findings suggest that during the COVID-19 pandemic, the crisis communication strategies employed by Chinese media on YouTube significantly increased the public's positive sentiments. Among them, enhancing strategies was applied most frequently and proved to be the most effective. (Introduction, paragraph five)

We earnestly appreciate your work and sincerely hope that our revisions meet your expectations. Again, thank you very much for your insightful comments and constructive suggestions.